# Strategies for Immunomonitoring after Vaccination and during Infection

**DOI:** 10.3390/vaccines9040365

**Published:** 2021-04-09

**Authors:** Lucille Adam, Pierre Rosenbaum, Olivia Bonduelle, Behazine Combadière

**Affiliations:** Inserm, Centre d’Immunologie et des Maladies Infectieuses, Sorbonne Université, 75013 Paris, France; lucille.adam@inserm.fr (L.A.); pierosenbaum@gmail.com (P.R.); olivia.bonduelle@sorbonne-universite.fr (O.B.)

**Keywords:** immune responses, vaccination, systems biology, immunomonitoring, COVID-19

## Abstract

Immunomonitoring is the study of an individual’s immune responses over the course of vaccination or infection. In the infectious context, exploring the innate and adaptive immune responses will help to investigate their contribution to viral control or toxicity. After vaccination, immunomonitoring of the correlate(s) and surrogate(s) of protection is a major asset for measuring vaccine immune efficacy. Conventional immunomonitoring methods include antibody-based technologies that are easy to use. However, promising sensitive high-throughput technologies allowed the emergence of holistic approaches. This raises the question of data integration methods and tools. These approaches allow us to increase our knowledge on immune mechanisms as well as the identification of key effectors of the immune response. However, the depiction of relevant findings requires a well-rounded consideration beforehand about the hypotheses, conception, organization and objectives of the immunomonitoring. Therefore, well-standardized and comprehensive studies fuel insight to design more efficient, rationale-based vaccines and therapeutics to fight against infectious diseases. Hence, we will illustrate this review with examples of the immunomonitoring approaches used during vaccination and the COVID-19 pandemic.

## 1. Introduction

Outbreaks of emergent infectious diseases, such as Ebola, Zika or coronavirus disease 2019 (COVID-19), are major healthcare issues with psychological, economical and demographic consequences for human societies. The handling of these numerous and recurring infectious threats, also including Human Immunodeficiency Virus (HIV), tuberculosis and malaria, illustrates that we still lack the knowledge, coordination and preparation to face those ordeals [1]. As a prestigious inheritance from forerunners such as Edward Jenner and Louis Pasteur, vaccination has, for a long time, proven to be an unrivaled weapon to protect populations against these threats [2]. Vaccination contributes to control infectious agents spreading among the population and provides individual and collective protection. Indeed, the success of mass immunization eradicated smallpox in 1980, a dreadful disease with a mortality rate of up to 50% and frequently bringing serious physical sequels in the survivors [3]. Vaccination also contributed to dramatically circumvent rabies, poliomyelitis, measles, mumps and rubella [4]. Other diseases induced by telluric bacteria such as tetanus became extremely rare, with mostly sporadic cases in the elderly with decreased immunity due to the lack of a vaccination booster [5]. Vaccination mimics natural infection, which will drive the organism to elicit a suitable memory immune response against the targeted pathogen, protecting against further infection. Consequently, immune surveillance after vaccination is necessary to assess vaccine efficacy. More generally, to suitably develop vaccines and discover insights about protective countermeasures, there is a need to track immune responses which contribute to protection or control of infections and translate them for vaccine development [6].

In vaccine development, one major challenge is to assess the relevant correlate(s) of protection (Figure 1) [7] consisting of biological markers that are directly associated with protection of the vaccinated individuals against the disease [8]. The complexity and heterogeneity of the immune response can hinder the identification and validation of these correlates. It is therefore essential to track modulations of the immune system through a process called immune monitoring (immunomonitoring). Immunomonitoring consists of a set of methods and assays allowing to measure or predict immune parameters from molecular to cellular level, but also to picture the nature and status of immune responses after infection or vaccination. The evaluation of correlates of protection mostly relies on humoral immune response induction through antigen-specific antibody titer and function measurements. Enzyme-linked immunosorbent assay (ELISA), neutralization, hemagglutination inhibition (HI; influenza) or opsonization assays (pneumococcus) are the most widely used methods to evaluate these humoral responses. Furthermore, antigen-specific CD4+ and CD8+ T immune response by enzyme-linked immunospot (ELISpot) or flow cytometry can be of major importance due to their complementarity with humoral responses [8]. The recent emergence of immune effector screening through complex approaches including multiparametric cell phenotyping (mass or spectral cytometry), transcriptomic, metabolomic, microbiome and proteomic analyses should lead to the discovery of new immune signatures associated with protection. These omics strategies generate large datasets which require powerful informatic and statistic tools. Machine learning would allow us to depict an overview of the immune response and provide insights about the major effectors of the immune response by systems biology-derived approaches [9]. This consists of a holistic approach aiming to characterize the immune response in its entirety, which opposes the mechanistic approaches dissecting a restricting amount of parameters [10]. Systems biology has led to a major change in how research is performed in many laboratories using these high-throughput and high-bandwidth methods, leading to the discovery of new relationships and participants in immunology, and generating new hypotheses about mechanistic processes [10]. Altogether, proper immunomonitoring should provide (1) insights about the protective parameters of the immune system and potential correlates of protection, (2) knowledge leverage of the mechanism of immunity after vaccination/infection, (3) insight into the persistence of immune memory and (4) the identification of early biomarkers predictive of immune efficacy for protection.

In this review, we discuss the available methods for tracking and identifying immune parameters that are involved in protection and the related predictors of immune responsiveness.

## 2. Identification and Measurement of Correlates and Surrogates of Protection

In the context of most natural infections, an efficient response would lead to the resolution of the disease with clearance of the infectious agents. In some cases, such as tuberculosis, immune responses can contribute to a reduction of symptoms in the absence of pathogen clearance [11]. In the context of vaccination, an efficient response would be protective against the infection (sterilizing protection) or at least prevent clinical severity.

Measurements of the correlates of protection are essential to evaluate vaccine efficacy, but their definition is still debated. According to Stanley A. Plotkin, correlates of protection have been defined as “an immune function responsible for and statistically interrelated with protection” [8]. He also specifies different degrees of correlates: an absolute correlate corresponds to “a specific level of response highly correlated with protection”, meaning that a quantitative threshold has been defined; a relative correlate corresponds to “a level of response variably correlated with protection”. Surrogates are defined as “an immune response that substitutes for the true immunologic correlate of protection, which may be unknown or not easily measurable”.

Identification of the correlates of protection is mostly based on immune response measurement, associated with the proportion of infection in the population compared with the control group [12]. Essential information can also be obtained from human challenge studies and from animal challenge models—with the caution that the mechanism of protection can differ between species. Similarly, rates of infection following passive specific antibody administration or according to mother-to-child antibody decline give information about the level of protective antibodies required for an infectious disease [13]. The correlates of protection might be relevant to subgroup individuals according to age, immune status (immunodepression, chronic disease…), genetic and environmental factors. For example, following influenza vaccination, the frequency and magnitude of the hemagglutinin inhibition (HI) titers remain lower in subjects over 65 years old than in younger adults [14]. However, influenza vaccination in the elderly still reduces the number of severe cases [15]. Thus, it is necessary to evaluate, in more detail, the immune parameters which contribute to protecting against influenza illness.

The most widely used correlate of protection is pathogen-specific antibodies produced after immunization. They strongly contribute to prevent further infection in many ways such as blocking pathogen entry into cells, immune complex formation, complement activation, opsonization enhancement, or complement- and antibody-dependent cellular cytotoxicity (Figure 1) [16,17,18]. These correlates are found to be well identified for the most encapsulated bacteria (*Haemophilus influenzae*, *Streptococcus pneumoniae* and *Neisseria meningitidis*) [19,20,21] or for toxin-producing bacteria, such as *Corynebacterium diphtheriae* and *Clostridium tetani* [22,23]. In the latter case, it is straightforward to see that functional antibodies directed against toxins—at the origin of the physiopathology—would be directly related to protection.

However, the identification of protective immune markers can be more complicated. For instance, antibodies against *Bordetella pertussis* toxin, fimbriae agglutinogens and pertactin are all involved without being fully responsible for protection, leading to difficulties in finding a consensus to properly assess a vaccine’s efficacy [7,24]. In the case of seasonal influenza, since Hobson et al. in 1972 [25] performed a viral challenge using live attenuated viruses to define protective antibody titers, this humoral immune response has remained the only parameter evaluated and standardized. HI measurement has been successfully used as a correlate of protection. Indeed, a 1:40 titer is currently considered as an immunologic correlate corresponding to a 50% reduction in the risk of developing influenza illness in adults [25,26]. Concomitant with the humoral response, the T cell responses appear to play a crucial role in protection against disease induced by subsequent infection with heterologous strains. Indeed, McMichael et al. demonstrated, in an attenuated influenza virus human challenge study, that CD8+ T cell responses were strongly participating in virus clearance, even in the absence of cross-reactive antibodies [27]. As a result, taking -mediated immunity into account cell in addition to antibodies is generally necessary to reach very high associations with protection [8].

In other contexts such as *Mycobacterium tuberculosis*, the lack of reliable correlates of protection remains a major impediment to vaccine development [28]. Indeed, this intracellular pathogen requires the mobilization of cellular responses. Interferon (IFN)γ-producing CD4+ T cells, cytotoxic T lymphocytes (CTL), and macrophages contribute together to the protective response, leading to granuloma formation [28,29]. In HIV infection, the lack of properly identified correlates of protection, despite intensive research effort, greatly complicates vaccine development. Indeed, it requires us to track neutralizing antibodies targeting diverse epitopes, and also to consider CD4+ T and CD8+ T cell responses [30]. For *Plasmodium falciparum*, inconsistent results for defining a universal correlate of protection led recent studies to stratify populations, taking age or malaria exposure into account [31]. For many diseases, we do not know which arms of the immune system are responsible for conferring protection, e.g., humoral versus cellular immunity, or whether systemic or mucosal immunity should be induced for sterilizing protection. Parameters for the maintenance of protective immunity over the years have also yet to be elucidated.

The emergence of the new SARS-CoV-2-induced coronavirus disease 2019 (COVID-19) outbreak illustrates the importance of identifying correlates of protection for rapid vaccine development. Indeed, this disease has rapidly emerged as one of the most important infectious insults of the modern era due to its extremely fast worldwide spread and severity. In March 2021, almost 120 million cases and more than 2.6 million deaths have been announced (from https://covid19.who.int/ website, accessed on 8 March 2021). Humoral response studies in SARS-CoV-2-infected patients rapidly revealed that infection successfully induces robust responses with detectable IgG, IgM and IgA antibodies (Table 1). Furthermore, anti-SARS-CoV-2 spike protein antibodies—more specifically against the receptor binding domain (RBD) region—are able to block virus entry into ACE-2 and CD147- expressing cells in vitro, demonstrating their neutralizing activity [32]. The neutralizing activity of SARS-CoV-2-induced antibodies was also demonstrated in vivo thanks to improvement in the condition of severely ill COVID-19 patients that received passive transfer of specific antibodies from recovered patients [33]. These data suggest that anti-SARS-CoV-2 antibodies directed against the RBD of the spike protein could be a relevant correlate of protection. These results were therefore essential for the SARS-CoV-2 vaccination field and revealed that vaccines designed to induce antibodies against the SARS-CoV-2 spike protein could be protective. According to this information, most of the SARS-CoV-2 vaccines available or under clinical trial evaluation are based on the viral surface spike protein and aim to provide sterilizing protection [34]. Despite the urgent need for a SARS-CoV-2 vaccine, lessons from the past arose about a potential harmful vaccine-induced immune response. Indeed, the vaccination field can count several examples where vaccines elicited increased infection or more severe disease in vaccinees compared with individuals in the control group [35]. This phenomenon, called immune enhancement, was observed with the HIV vaccine candidate based on human adenovirus 5 vector from the STEP study and led to the emergency arrest of the trial [36]. Immune enhancement, leading to more severe diseases in vaccinated or infected people, was also observed with dengue virus, respiratory syncytial virus (RSV) and severe acute respiratory syndrome (SARS) induced by SARS-CoV. One mechanism leading to this unwanted vaccine-induced immune response is antibody-dependent enhancement (ADE), which facilitates the infection of cells through uptake via macrophages [35]. ADE was indeed confirmed in some cases of secondary infection with dengue virus. Besides, some vaccines eliciting a Th2-type response might promote allergic inflammation and poorly functional antibody response, which can eventually lead to immune complexes formation and complement activation, resulting in immunopathology. This mechanism was suspected to increase disease severity in infants vaccinated with a RSV vaccine candidate [37]. In the case of SARS-CoV-2, the concern about ADE or a harmful Th2-type response was a matter of interest during vaccine development. However, the extensive characterization of the SARS-CoV-2 adaptive immune response revealed neither evidence of ADE nor a Th2-oriented response in COVID-19 patients. Finally, the results of the first clinical trials show protective results and are reassuring concerning the safety of vaccine-induced immune responses in vaccinees. Otherwise, the very promising results of the Phase III clinical trials involving RNA-based vaccines demonstrated up to 95% protection, highlighting a major relationship between the anti-spike humoral response and protection (Table 2) [38]. These results tend to be confirmed in Phase IV mass vaccination programs [39]. However, the question regarding the maintenance of this level of protection in the long term needs to be addressed in the future.

## 3. Immunomonitoring of Correlates and Surrogates of Protection

Two major arms of immunity are currently measured during vaccination and after infection, defined as innate and adaptive immunity. These two arms closely interact together to modulate the immune response, leading to variable degrees of protection against an infectious disease [52,53]. The early innate immune response occurs from hours to days after infection or vaccination, and differs according to antigen recognition and vaccine preparation (antigen design, adjuvant, vector…). Non-self components (pathogens, vaccines) are sensed by local cells through pattern recognition receptors (PRR) such as the Toll-like receptor (TLR), the Nod-like receptor (NLR), retinoic acid-inducible gene (RIG)-1-like helicase, the scavenger receptor or the mannose receptor [54]. The activation of these receptors results in the induction of an inflammatory signal (like cytokines and chemokines) that triggers recruitment of various immune cells such as dendritic cells, monocytes, neutrophils and natural killer cells to the site of infection/vaccination. These cells, more particularly, antigen-presenting cells, contribute to the antigen capture and transport in secondary lymphoid organs to induce the different T and B effector cells of adaptive immunity. Naïve CD4+ and CD8+ T cells recognize the presented antigen and go through clonal expansion and differentiation. CD4+T cells differentiate into T follicular helpers (Tfh), regulatory T cells (Treg) or other effectors (Th1, Th2, Th17) specializing in cytokine production. Depending on their differentiation, these cells support CD8 T cell activation and differentiation into CTL and B cell activation, or participate in the regulation of the adaptive response through inhibitory signals. The differentiation state of T cells is characterized by a specific phenotype that can be tracked. One of the most classical combination markers used to define T cell differentiation is the surface expression of CD45RA, CD45RO and CCR7. For example, naïve cells can be defined as CD45RA+CD45RO-CCR7+ while effector memory cells are defined as CD45RA-CD45RO+ and CCR7- and central memory cells are defined as CD45RA- CR45RO+ CCR7+ [55].

B cells, which are the source of antibodies, contribute importantly to the protection against infection provided by vaccines. These cells recognize antigens through their B cell receptor (BCR). Once activated, B cells can undergo immunoglobulin (Ig) class switch recombination and become antibody-secreting plasma cells. Therefore, mature B cells express IgM and IgD, while class-switched B cells express IgA (essential for mucosal protection), IgE or IgG. To further gain in specificity, B cells can, in their germinal center and in a Tfh cell-dependent manner, go through hypersomatic mutation. The activation state of B cells can also be tracked according to their phenotype (plasma cells express CD38, CD138, CD27 and a low level of CD20; memory B cells express CD27 but lack plasma cell marker expression) [56].

Immunomonitoring of innate and adaptive events therefore appear essential to finely characterize immune responses leading to protection and vaccine efficacy. Therefore, the accurate measurement of variations in the human immune system requires precise and standardized assays to distinguish true biological changes from technical artefacts [57]. Moreover, the vaccine type, dose and adjuvants can affect the immunomonitoring strategy by inducing different immune response profiles and kinetics [58,59]. For instance, live attenuated vaccines mimic natural infection, generally yielding a broad and long-lasting immune response. At the opposite end, inactivated and subunit vaccines are often less immunogenic at equivalent doses [60,61]. Indeed, some vaccines are designed to induce a pathogen-specific antibody response due to their efficacy to fight diphtheria, tetanus, polio or, more recently, SARS-CoV-2. The parameters leading to a protective humoral response (level of antibodies, neutralizing activity…) need to be taken into account in the choice of the immunomonitoring methods (ELISA, neutralizing assay) [53]. Additionally, some vaccines, usually based on viral vector or nucleic acids (DNA [62], RNA [63]), for example, are specifically designed to induce cellular immunity [64]. In this case, control of the pathogen load at low levels and the reduction of transmission risk are aimed rather than sterilizing immunity. This type of response appeared particularly interesting in the HIV vaccination field when the induction of sterilizing immunity through antibodies remained a challenge. Immunomonitoring of a cellular-specific response is therefore required and is usually performed through ELISpot or flow cytometry.

In addition, the emergence a rationally-based vaccine in the near future [65] would lead to better knowledge of the immune effector triggered. As a result, some immunomonitoring methods also need to be adapted accordingly.

### 3.1. Pathogen-Specific Easy-to-Use Assays

Well-known antibody-based techniques such as ELISA or ELISpot have been used for decades, especially in the vaccine development field. Based on highly specific and validated antibodies, these methods are a powerful tool for solubilized protein detection and quantification from a large range of samples, with the required quantity of sample per assay usually being small. These methods have the advantages of being simple to perform and easy to analyze [66,67]. Nowadays, efforts have been made to improve the sensitivity of these technologies—such as digital ELISA—or increase the number of analytes evaluated [68]. Alternatively, assessing the correlates of protection or surrogate markers often relies on functional assays, particularly neutralizing antibody measurement such as the Plaque Reduction Neutralization Test or, more recently, the fluorescence-based neutralization assay [69]. In this context, exploration of other pathogen-specific antibody functions can also be relevant such as HI activity, opsonophagocytic activity [70] or complement fixation properties [71]. Carrying out more extensive studies associated with pathogen-specific antibodies and assessing the affinity, targeted epitopes, Ig isotypes and subclasses may also allow us to better picture their mechanism of action. Despite these methods being well standardized and considered as references for immune response evaluation, they often are plagued by their lack of reproducibility. Inconsistency could be also due to sample handling, the reagents used, protocol modifications, inter-operator bias or device settings. To circumvent reproducibility problems, more and more effort has to be made in order to improve protocol standardization. For example, automates or premade tubes containing reagents have been developed to limit inter-operator bias [72].

### 3.2. Immune Cell Phenotypic and Functional Analysis

In the past decade, flow cytometry has emerged as a valuable tool for phenotypic and functional analysis of immune cells from whole blood, peripheral blood mononuclear cells (PBMCs), and tissue samples (lung, bronchoalveolar lavage, spleen, skin, liver…). Flow cytometry analysis would allow an overview of cell dynamics and activation state phenotypes. Multiple markers and strategies are currently available [73,74] to discriminate the differentiation and activation states of CD8+ and CD4+T cells through the use of markers such as CCR7, CD45RA, CD25, CD95 or CD27. Flow cytometry is also routinely used to assess pathogen-specific T cell responses through intracellular cytokine staining (ICS), usually measuring interleukin (IL)-2, IFNγ, tumor necrosis factor (TNF)α or IL17. The evaluation of T cells response through cytokine-independent assays [75] has also been successfully used to evaluate the T cell response after vaccination in a range of studies [76,77,78]. These T cell receptor (TCR)-dependent activation-induced marker (AIM) assays that measure the expression of a combination of activation markers (CD69, OX40, CD137…) at the T cell surface have permitted a fast and precise evaluation of the frequency of T cell response in the context of the SARS-CoV-2 pandemic [79].

Similar strategies also allow us to accurately characterize B cell subsets, including immature, naïve, class-switch recombined and plasmocytes, using cell surface markers such as HLADR, CD19, CD20, IgD, IgM, IgG, CD25, CD138, CD27 and CD38 [80,81]. To sum up, this technology has the advantage of simultaneously characterizing multiple intracellular and/or extracellular markers per cells, at the expense of expertise in panel design and the analysis of such data. The flexibility of this system has made it extremely widely exploited, but data obtain remain complex to harmonize in multicentric studies [82]. Additionally, the number of simultaneously usable markers remains limited.

To circumvent this limitation, recent technological breakthroughs have led to the emergence of high-multiparametric cytometry, allowing the simultaneous evaluation of an ever-growing amount of markers per cell (up to 50 markers to date). High-dimensional time-of-flight mass cytometry (CyTOF) consists of a combination of flow cytometry and mass spectrometry, where antibodies are conjugated to rare heavy metal isotopes instead of fluorochrome. After cell staining, the sample is ionized on a plasma torch and the time of flight of these rare isotopes allow the reconstitution of individual cell marker’s expression. This technology also has the advantage of an important spill-over reduction in comparison with flow cytometry. Cell population characterization by CyTOF is nowadays more and more widespread and allows an accuracy level in immune response dissection that was not reached with conventional flow cytometry [83]. Alternatively, spectral flow cytometry is the evolution of conventional flow cytometry. However, the system of signal detection does not rely on photomultiplier tubes, which collect a fraction of the fluorophore emission spectrum, but rather on the collection, analysis and recognition of the full emission spectrum, used as a reference in multicolor applications [84]. This technology allows the simultaneous use of fluorophores with very close emission spectra such as allophycocyanin (APC) and Alexa Fluor 647. These improvements have led to more resolutive results than regular flow cytometry [85]. Such new technologies allow an in-depth characterization of the cell phenotype, function and activation state, which is particularly interesting in the context of vaccine or infectious disease monitoring. These highly multiparametric methods require bioinformatic expertise, allowing unsupervised analysis [86], with the advantage of automatically defining cell subsets that might been not highlighted with conventional supervised analysis, in addition to getting rid of operator-driven analysis bias. This may lead to the identification of novel cell populations that could be associated with an effective vaccine response [87,88].

## 4. High-Throughput Immunomonitoring Techniques

Important technological innovations have been made in the past years in flow cytometry and other systems devoted to tracking immune responses such as multiple cytokine measurement devices [89]. However, these technologies, taken individually, are not self-sufficient to recapitulate the immune system’s complexity.

Concomitantly with development of next-generation sequencing (NGS) platforms, high-throughput devices have also become more and more sensitive and affordable [90], and allow an indiscriminate analysis of thousands to millions of parameters such as gene sequences, transcripts, metabolites, proteins, lipids or polysaccharides. These very sensitive technologies could contribute to the identification of potential surrogate markers of immunogenicity or protection after vaccination and disease outcome after infection.

More particularly, the revolution of single-cell RNA sequencing (scRNA-seq) technologies has led to a better understanding of the diversity and development of immune cells by multiple gene expression analysis. In 2009, the first study using sequencing at the unicellular level allowed the characterization of cells at the early stage of development [91]. Over past few years, scRNA-seq technologies have developed from a handful of individual cells to hundreds of thousands in a single experiment with sensitivity and accuracy despite a very small amount of biologic material [92]. The principle of scRNA-seq consists of isolating cells and performing RNA reverse transcription followed by cDNA amplification and library preparation before NGS. Single-cell isolation was achieved through the development of flow-cell sorting and microfluidic technologies (for example Fluidigm C1 limited for up 800 cells per chip and a homogeneous cell size, or microdroplet-based fluidics like the Chromium system from 10X Genomics that could screen thousands to millions of barcoded cells in microdroplets). During the last decade, a lot of multimodal single-cell measurements were developed to associate the transcriptomics, repertoire, epigenomics, proteomics and genomics of immune single-cells [93]. NGS also allowed us to simplify TCR and BCR repertoire evaluation through the Rep-Seq technique [94]. For instance, BCR profiling led to a better understanding of an alternative to an isotypic switch in the B cell (locus suicide recombination), resulting in the deletion of a constant region that could restrain the activation of mature B cells [95]. Technologies depicting the Ig repertoire and evaluating the redundancy of clonal sequences have been found to be valuable in multiorgan diseases [96] but also in vaccination [97]. All these technologies are a real opportunity to characterize small cell populations and gene pathway implications, and to decipher the immune response in infectious disease and vaccines [98].

## 5. Systems Biology

New immunomonitoring techniques have paved the way for the current systems biology era. Opposite to dissecting isolated immune mechanisms or functions, systems biology aims to take advantage of multiparametric datasets to provide new insights about the general behavior of the immune system. It has become an essential feature of identifying predictive innate biomarkers.

Indeed, components of the adaptive response that are widely used as correlates of protection to assess vaccine efficacy require days to weeks to be acquired. In the context of a pandemic, as the world is facing now with SARS-CoV-2, the population needs to be urgently protected through vaccination. The early evaluation of vaccine efficacy, to predict vaccine responders from non-responders, can be extremely relevant. In this context, the idea is to establish correlations between one or several early-induced non-antigen-specific biomarkers—an innate immune signature—with highly specific putative correlates of protection. Nevertheless, many early vaccine signatures associated with adaptive response parameters have been proposed and constitute innate biomarker candidates. Yellow fever vaccine (YF-17D), one of the most effective vaccines available, has been widely studied to identify these early signatures and dissect the relationship between the innate and adaptive response [99,100,101,102,103]. Molecules involved in stress response pathways, i.e., EIF2K (also called GCN2), and the complement protein (C1qB) have been shown to predict, with up to 90% accuracy, the CD8+ T cell responses, while *TNFR17* predicted the neutralizing antibody response with up to 100% accuracy [102]. These early signatures were able to accurately predict the adaptive response outcome measured 2 months following vaccination. In the context of influenza vaccination with the trivalent influenza vaccine, we demonstrated that a minimal gene signature of nine genes associated with the serum CXCL10 level measured as soon as day1 after vaccination was able to predict specific granzyme B-producing CD8+ T cells and antibody responses 21 days later [5].

This kind of holistic approach requires the use of bioinformatic tools for data integration such as dimensionality reduction visualization t-SNE, UMAP, PCA, MDS…), and analyzing algorithms (LDA, LASSO…) [102,104]. Omics studies have already highlighted the role of very diverse parameters interfering with vaccine-induced immunity, such as endocrine hormones [105], the microbiota population [106] and nutrient receptors [107]. Systems biology would be probably a cornerstone of rationale-based vaccine development, with computational biology tools allowing one to select, in silico, immunogenic T cell epitopes that are more susceptible to inducing an adequate immune response [108]. Along with immune parameters, computational tools can also take intrinsic host characteristics into account, such as HLA profiling, which significantly contribute to genetic susceptibility to infectious diseases and variations in the response to vaccines [109]. Other host-related factors also influence immune responses, such as age, gender, social environment or chronical diseases. The integration of the different data levels using omics technologies and bioinformatical modeling can depict the strength of the immune response to vaccines over time from the molecular networking to the cellular composition and cell-to-cell cross-talk following vaccination, and paves the way forward to the personalized vaccination concept [109], where all vaccine doses would be adapted based on each subject’s history.

In the context of COVID-19 disease, important discoveries about the SARS-CoV-2-induced immune response have been made possible thanks to cutting-edge technologies based on integrative approaches. Indeed, a very large and heterogenic combination of immunomonitoring techniques—shortly exemplified in Table 1 and Table 2—succeeded in providing insights about how to fight against this disease.

## 6. Conclusions

In the past years, major advances in immunomonitoring methods have seen the promise of systems biology era. Slowly, integrative data strategies taking advantage of high-throughput devices have depicted, more and more precisely and rapidly, the immune processes associated with vaccination, infection or cancers. These technologies have already begun to provide insights about the predictive biomarkers or immune mechanisms triggered by diseases such as COVID-19. As a counterpart, it demands important amounts of time and resources, including interdisciplinary collaboration among mathematicians, informaticians, immunologists, and also chemists and physicians. International partnerships, consensual standard operating procedures, biobanking, protocols and database management are still essential to establishing relevant data integration and meta-analysis. Undoubtably, new discoveries resulting from this continuously arising holistic field will lead to more rationale-based vaccine development. The incredibly fast development of vaccines to fight against the COVID-19 pandemic has demonstrated the incredible mobilization capacity of industrial and academic researchers in a sanitary crisis context. It brings hope that a highly effective, collaborative and coordinated worldwide research taskforce could be established to face future pandemics.

## Figures and Tables

**Figure 1 vaccines-09-00365-f001:**
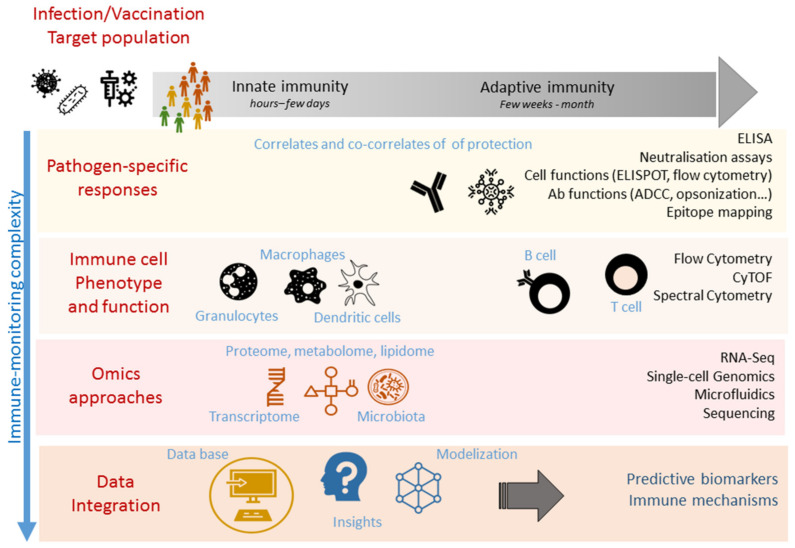
Immunomonitoring techniques for vaccination. Immunomonitoring of innate and adaptive immune responses is based on a collection of biological samples, either freshly collected or cryoconserved, which have been ob-tained after infection or vaccination. Various immunomonitoring approaches are summarized, from easy-to use methods such as ELISA or ELISpot, to more complex methods such as CyTOF, spectral cytometry or RNA-seq that depict more precise fea-tures of the immune responses. However, these methods require skill in informatics and statistics.

**Table 1 vaccines-09-00365-t001:** Overview of the immunomonitoring approaches used in SARS-CoV-2 infection.

References	Patient Cohort	Findings	Immunitoring Techniques
Hadjaj et al. [40]	18 Healthy Donors 15 Mild 17 Severe 18 Critical	Type I IFN impairment Exacerbated inflammatory response	Mass Cytometry mRNA Expression Analysis Multiplex Cytokine Detection
Combadiere et al. [41]	38 Critical	Myelemia with overabundance of CD123+ and LOX-1+ neutrophils	Flow Cytometry Ultra-sensitive Digital Immunoassay (Quanterix)
Weiskopf et al. [42]	10 Severe/Critical	Immunomodulation of T-cell responses depending on severity	Flow Cytometry ELISA Multiplex Cytokine Detection
Laing et al. [43]	55 Health Donors 56 Severe/Critical	CXCL10, IL-10, IL-6; B and T, and monocyte subset signatures related with severity	Flow Cytometry mRNA Expression Analysis Multiplex Cytokine
Wen et al. [44]	10 Recovering COVID-19 Patients	SARS-CoV2-specific IGHV3-23-IGHJ4 pairing TNFSF13, IL-18, IL-2, and IL-4 genes may benefit from COVID-19 recovery	TCR and BCR Sequencing Single-Cell RNA-Seq
Silvin et al. [45]	72 Healthy Donors 27 Mild 16 Moderate 43 Severe	Non-classical monocytes and calprotectin-producing immature neutrophils increase in severe cases	Spectral Cytometry Mass Cytometry Flow Cytometry Single-Cell RNA-Seq Multiplex Cytokine Detection

IFN, interferon; ELISA, enzyme-linked immunosorbent assay; TCR, T cell receptor; BCR, B cell receptor.

**Table 2 vaccines-09-00365-t002:** Overview of the immunomonitoring approaches used in SARS-CoV-2 vaccine clinical trials.

References Clinical Trial ID	Phase Patient Cohort	Vaccine	Immunitoring Techniques
Jackon et al. [46] NCT04283461	Phase I 45 healthy adults Age: 18–55	*Moderna vaccine* RNA-based vaccine mRNA-1273 → Spike Dose escalation (25 μg, 100 μg, 250 μg) Homologous prime boost	ELISA Neutralization assay ICS—flow cytometry
Keech et al. [47] NCT04368988	Phase I–II 132 healthy adults Age: 18–59	*Novavax vaccine* Protein-based vaccine NVX-CoV2373 → Spike with/without Matrix-M1 adjuvant dose escalation (5–25 ug) Homologous prime boost	ELISA microneutralization assay ICS—flow cytometry
Logunov et al. [48] NCT04436471 NCT04437875	Phase I–II 120 healthy adults Age 18–60	*Sputnik V vaccine* Viral vector-based vaccine rAd26 and rAd5 → spike Heterologous prime boost Prime rAd26-S Boost rAd5	ELISA Neutralization assay Proliferation assay
Mulligan et al. [49] NCT04368728	Phase I–II 45 healthy adults Age 18–55	*Pfizer-BioNTech vaccine* RNA-based vaccine BNT162b1→ Lipid nanoparticle-formulated nucleoside-modified mRNA vaccine Trimerized SARS-CoV-2 RBD Dose escalation: 10 μg–30 μg–100 μg Homologous prime boost	IgG binding assay Neutralization assay
Folegatti et al. [50] NCT04324606	Phase I–II 1077 adults Age: 18-55	*Astrazeneca vaccine* Viral vector-based vaccine ChAdOx1 nCoV-19 → spike Homologous prime boost	ELISA Neutralization assay ELISpot
Zhang et al. [51] NCT04352608	Phase I 143 healthy adults Phase II 600 healthy adults Age: 18–59	*Sinovac vaccine* Inactivated virus-based vaccine CoronaVac → inactivated SARS-CoV-2 Dose escalation	ELISA Microcytophathogenic effect assay ELISpot

ELISA, enzyme-linked immunosorbent assay; ELISpot, enzyme-linked immunospot; ICS, intracellular cytokine staining.

## Data Availability

Not applicable.

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
