# Peer review of "Strategies for Immunomonitoring after Vaccination and during Infection"

_vaccines, 2021, doi:10.3390/vaccines9040365_

Round 1
Reviewer 1 Report
- These authors discuss immunomonitoring which they define as the study of an individual's immune response over the course of vaccination or infection. During infections these efforts involve studying innate and adaptive immune responses. After vaccination these efforts involve studying antibody responses and cellular responses.
- This review provides an in-depth and relatively complicated discussion of these ideas. In some places word choice seems a little peculiar. For example, "vaccination acts as a lure”, " furthermore, "infer about antigen specific methods ", various substances are "intricated". The text might be reviewed to eliminate some of these choices.
- There has been some discussion during the development of Covid vaccines that vaccination might result in harmful immune responses. Should those considerations be discussed at least briefly in this review?
4. The tables and figure probably need some foot notes to clarify abbreviations.
Author Response
Reviewer # 1:
Comments and Suggestions for Authors:
- These authors discuss immunomonitoring which they define as the study of an individual's immune response over the course of vaccination or infection. During infections these efforts involve studying innate and adaptive immune responses. After vaccination these efforts involve studying antibody responses and cellular responses.
- This review provides an in-depth and relatively complicated discussion of these ideas. In some places word choice seems a little peculiar. For example, "vaccination acts as a lure”, " furthermore, "infer about antigen specific methods ", various substances are "intricated". The text might be reviewed to eliminate some of these choices.
We simplified some terms and sentences in an attempt to clarify the discussion. We modified the concerned sentences to choose a more adequate wording.
P1: “Vaccination acts as a lure mimics natural infection,“
P2: “Furthermore, infer about antigen-specific CD4+ and CD8+ T immune response by Enzyme-Linked ImmunoSpot (ELISpot),”
P3: “For instance, antibodies against Bordetella pertussis toxin, fimbriae agglutinogens and pertactin are all intricated involved without being fully responsible of protection,”
- There has been some discussion during the development of Covid vaccines that vaccination might result in harmful immune responses. Should those considerations be discussed at least briefly in this review?
Thank you for this interesting remark. We discussed more into the details about the potential harmful responses induced after vaccination.
P4: “Despite the urgent need of a SARS-CoV-2 vaccine, lessons from the past arose concern about a potential harmful vaccine-induced immune response. Indeed the vaccination field count several examples were vaccines elicited increased infection in vaccinees compared to individuals in control group [35]. This phenomenon called immune enhancement was observed with the HIV vaccine candidate from Merck during the STEP study and lead to the emergency arrest of the trial [36]. Immune enhancement, leading to more severe diseases in vaccinated or infected people was also observed with dengue virus, Respiratory Syncytial Virus (RSV) or SARS. One mechanism leading to this unwanted vaccine-induced immune response is Antibody dependent enhancement (ADE), which facilitates infection of cells through uptake via macrophages [35]. ADE was indeed confirmed in some case of secondary infection with dengue virus. Besides, some vaccines eliciting Th2 type response might promote allergic inflammation and poorly functional antibody response, which can eventually lead to immune complex formation and complement activation resulting in immunopathology. This mechanism was suspected to increase disease severity in infants vaccinated with an RSV vaccine candidate [37]. In the case of SARS-CoV-2 the concern about ADE or harmful Th2 type response was a matter of interest during vaccine development. However the extensive characterization of SARS-CoV-2 adaptive immune response revealed neither evidence of ADE, nor Th2-oriented response in COVID-19 patients. Finally the results of first clinical trials show protective results and reassuring concerning the safety of vaccine induced immune responses in vaccinees.”
- The tables and figure probably need some foot notes to clarify abbreviations.
Footnotes have been added for both table and figure to clarify main abbreviations

Reviewer 2 Report
The authors summarized the methods included in “immunomonitoring”. The authors also provide evidence of the contributions of the immunomonitoring methods in the COVID-19 pandemic. It would be delightful if my comments some help to improve the manuscript.
MINOR COMMENTS
1. I think the “immunomonitoring” is still a not familiar idea for most readers. The clear explanation about the definition of the word seems to be needed.
2. p.3 l. 150. I agree the SARS-CoV-2 is a good subject about mentioning the relationship between vaccination and immunomonitoring. However, focusing on the ongoing matter, reducing the value of the review article, because the information becomes obsolete soon. I suppose the author could refer to more settled evidence.
3. Nowadays, there are several types of vaccines, such as peptide antigens, inactivated microbes, and RNA. It could affect choosing a suitable monitoring method in vaccine development. If possible, the author can organize and classified them.
Author Response
Reviewer # 2:
Comments and Suggestions for Authors:
The authors summarized the methods included in “immunomonitoring”. The authors also provide evidence of the contributions of the immunomonitoring methods in the COVID-19 pandemic. It would be delightful if my comments some help to improve the manuscript.
MINOR COMMENTS :
- I think the “immunomonitoring” is still a not familiar idea for most readers. The clear explanation about the definition of the word seems to be needed.
We thanks the reviewer for this comment. We now added an explanation of the term immunomonitoring in the introduction.
P2: “It is therefore essential to track modulations of the immune system through a process called immune monitoring (immunomonitoring). Immunomonitoring consists in a set of methods and assays allowing to measure or predict immune parameters from molecular to cellular level allowing to evaluate the nature and state of immune responses after infection or vaccination”
- 3 l. 150. I agree the SARS-CoV-2 is a good subject about mentioning the relationship between vaccination and immunomonitoring. However, focusing on the ongoing matter, reducing the value of the review article, because the information becomes obsolete soon. I suppose the author could refer to more settled evidence.
We understand indeed to Sars-CoV2 topic might be obsolete soon. However SARS-CoV-2 has only been used in this review to illustrate immunomonitoring techniques that can be used to track vaccine/infection immune response. Those methods would remain relevant in other context. Also, we extended the discussion to other infections depending on the topic (influenza, yellow fever for innate biomarkers, for instance)
P8: “Yellow fever vaccine (YF-17D), one of the most effective vaccines available, has been widely studied to identify these early signatures and dissect relation between innate and adaptive response [87–91]. Molecules involved in stress-responses pathway, i.e EIF2K (also called GCN2), and the complement protein C1qB has been shown to predict with up to 90% accuracy the CD8+ T cells responses while TNFR17 predicted the neutralizing anti-body response with up to 100% accuracy [90]. These early signatures were able to accurately predict adaptive response outcome measured 2 months following vaccination. In the context of influenza vaccination with the trivalent influenza vaccine, we demonstrated that a minimal gene signature of 9 genes associated with the serum level CXCL10 measured as soon as D1 post vaccination was able to predict specific granzyme B producing CD8+ T cells and antibody responses 21 days later [5].”
- Nowadays, there are several types of vaccines, such as peptide antigens, inactivated microbes, and RNA. It could affect choosing a suitable monitoring method in vaccine development. If possible, the author can organize and classified them.
We added a paragraph where we developed about vaccine type. It could indeed have an important impact on the choice of immunomonitoring methods used.
P5 : “Also, vaccine type, dose, as well as adjuvants, can affect immunomonitoring strategy by inducing different immune response profiles and kinetics [46,47]. For instance, live attenuated vaccine mimic the natural infection, generally yields broad and long-lasting immune response. At the opposite, inactivated and subunit vaccines are often less immuno-genic at equivalent doses [48,49]. Indeed some vaccines are designed to induce pathogen specific antibody response, due to their efficacy to fight diphtheria, tetanus, polio or more recently SARS-CoV-2. The parameters leading to protective humoral response (level of antibody, neutralizing activity…) need to be taken into account in the choice of the immunomonitoring methods (ELISA, neutralizing assay) [41]. Additionally, some vaccines, usually based on viral vector or nucleic acid (DNA [50], RNA [51]) for example, are specifically designed to induce cellular immunity [52]. In this case it is the control of the pathogen load at low level and the reduction of transmission risk that is aimed rather than sterilizing immunity. This type of response appeared particularly interesting in HIV vaccination field when the induction of sterilizing immunity through antibody remains a challenge. Immunomonitoring of cellular specific response is therefore required and usually performed through ELIspot or flow cytometry.
In addition, emergence rationally-based vaccine in the near future [53] would lead to a better knowledge of the immune effector triggered. As a result some immunotoring methods also need to be adapted accordingly.”
